# 'We Are Children of God': An Ethnography of a Catholic Community in Rural China in the COVID-19 Pandemic

**Wei Xiong** [1,*] and **Xinan Li** [2]

1    School of Chinese Language and Literature, Central China Normal University, Wuhan 430079, China
2    School of Social Sciences and Humanities, Loughborough University, Loughborough LE11 3TU, UK;
     x.li5@lboro.ac.uk
*    Correspondence: xiongwei@mail.ccnu.edu.cn

**Abstract:** Academic studies of the relationship between religion and pandemics have been emerging since the beginning of the COVID-19 pandemic. However, many of these studies have been conducted in Euro-American contexts, with little attention paid to non-Western cases. This article provides a local case study from China, the earliest epicenter of the pandemic. The study focused on a Catholic community in rural China, Little Rome, through the lens of lived religion, exploring the relationship between religion and the COVID-19 pandemic. Participants in our ethnographic study indicated that the Church plays an essential role in responding to the pandemic. In contrast to conventional studies of lived religion, in this ethnographic study on Catholicism in China, we contend that while the study of the lived experience of individuals is central to the lived-religion approach, more attention needs to be paid to the role of religious institutions such as the church, which mediate relations between individuals, society, and other social institutions. This article also argues that investigating different places and cultures can provide rich data for understanding the dynamic and diverse relationship between religion and the pandemic.

**Keywords:** Catholicism; lived religion; religious institutions; pandemic; ethnography; China

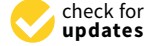

## 1. Background

The outbreak of the novel coronavirus in Wuhan, Hubei Province, China, in December 2019, which was later officially named COVID-19 by the World Health Organization (WHO), was declared a global pandemic on 11 March 2020.[1] This is a public health crisis that endangers all human society and occurred at the start of the second decade of the 21st century.

In the COVID-19 pandemic, mass gatherings have become one of the most dangerous factors in spreading the disease. Thus, social distancing has become one of the most effective ways to prevent the further spread of the virus (Ebrahim and Memish 2020). However, with mass gatherings being one of the most common activities in collective worship, religion seems to have facilitated the spread of COVID-19 in some circumstances. Initially, adherents of religions in many countries were concerned that the tightened social control policies implemented by the state would interfere with and contradict the principle of freedom of religion. This concern has caused some to violate government regulations in the pandemic, and their behavior has increased the risk of virus transmission (DeFranza et al. 2020). For instance, researchers have pointed out that the mass gatherings of the Shincheonji Church in South Korea were a crucial cause of mass contagion of COVID-19 among its adherents (Quadri 2020). Researchers from The Netherlands have found that religion is in part responsible for the direct and indirect spread of the virus through rigorous statistical analysis, and have called for more attention to be paid to the religious factor in the pandemic (Vermeer and Kregting 2020). However, social distancing, the most common and effective means of curbing the pandemic, has profoundly impacted and

changed the operational, organizational, and structural practices of many faith traditions, suggesting new directions in research on religion (Baker et al. 2020).

Religion is an essential factor that underlies human attitudes and actions in response to social crises (Bentzen 2019). Experiencing the enormous impact of the COVID-19 pandemic, many people have turned to religion for psychological and spiritual comfort (Bentzen 2020; Molteni et al. 2020; Boguszewski et al. 2020). In a Pew report (2020), statistics indicate that over half of the adults in the United States have prayed for the end of the COVID-19 pandemic.[2] Empirical research also provides evidence for the positive effects of religious faith on people's psychological health while facing the pandemic (Thomas and Barbato 2020; Isiko 2020). The literature also suggests that variables such as age, gender, rites, and frequencies of religious attendance can correlate with and affect religious belief and expression in the pandemic (Meza 2020). Apart from psychological and spiritual comfort, religious organizations and individuals have also participated in practical work in the pandemic. Many have acted in compliance with religious ethics and moral responsibilities, providing medical care for the sick and professional religious services, which may have led to the death of many Catholic priests in Italy (Bramstedt 2020).

Presently, much of the scholarly literature on religion and the COVID-19 pandemic has documented data and cases in Euro-American contexts; only a few studies have conducted research in other regions around the world. Many studies have employed a quantitative methodology, particularly statistical methods (e.g., Boguszewski et al. 2020; Meza 2020; Vermeer and Kregting 2020); qualitative, in-depth inquiries are scarce. This research seeks to bridge the gap by providing an ethnographic account of a non-Euro-American case in the Global East, specifically China.

In the West, religion presents a resurging trend in the public arena (Reynolds 2015). Christianity, despite the contention between secularization and sacralization (Nyhagen 2017), has been an influential factor in people's everyday lives, which may have positively and negatively contributed to the containment and prevention of the pandemic. However, due to the Chinese Communist Party-state's widespread repression and restriction of religion, it is almost invisible in Chinese public media discourse and other areas of the Chinese public arena. Moreover, the pandemic seems to have empowered the Chinese state's authority in its institution of top-down social control (Zhou 2020). In this context, the survival of religious organizations in China largely depends on government policies (Qiang and Lu 2020), exhibiting limited social functions (Hu and Sidel 2020).

To better understand the operation, interaction, and practice of religion in China in the pandemic, this ethnographic study sought to transcend existing approaches and analyze the subject through the lens of lived religion (e.g., Ammerman 2006, 2020). Returning to the level of everyday life and observing the experiences, emotions, and practices of rural Catholic communities in China during the pandemic, this analysis aimed to deepen the understanding of lived religion and capture the culture and reality of the "survival" and "revival" of religion in contemporary China (cf. Yang 2012).

## 2. Theoretical Perspectives

In recent years, scholars have advocated the approach of lived religion, suggesting that more scholarly attention needs to be paid to 'how religion and spirituality are practiced, experienced, and expressed by ordinary people (rather than official spokespersons) in the context of their everyday lives' (McGuire 2008, p. 12). This approach has offered an alternative to the conventional institution-centered theory and methodology of research on religion (e.g., Orsi 2003; Ammerman 2006, 2016, 2020; Neitz 2011). The lived-religion approach challenges many binaries in the scholarly study of religion, such as public and private, political and domestic, religious and secular, imagination and reality, and individual and institutional (Orsi 2003; Ammerman 2006, p. 9).

The emergence of lived religion studies is related to dissatisfaction with secularization and rational choice theories. Scholars invoke practical theories and focus on the choices,

agency, and practices of religious individuals. Along this study path, the role of officials and institutions in religious activity naturally diminishes (Edgell 2012). Many scholars consider lived religion as a rejection of officially sanctioned 'proper' religion (Hall 1997, pp. vii–xiii) and exclude institutional factors (McGuire 2008, p. 12). This view is related to the Western neoliberal concept of individualism (Ammerman 2020).

Many scholars argue that focusing solely on individuals and unofficial religion does not reflect an accurate picture of lived religion (Repstad 2019, pp. 55–66). Religion is 'not necessarily private or internal'; 'it is often practiced in public or in collective acts and understandings' (Neitz 2011, p. 54). Thus, there is a dynamic relationship between religious institutions and individuals (McGuire 2008, pp. 4–5). We argue that research should focus on both official and more fluid, unofficial religious expressions (Orsi 2003); religious institutions need to be included in the analysis of lived religion (Edgell 2012; Ammerman 2016; Vejrup Nielsen and Johansen 2019).

Due to the predominant Marxist-atheist ideology in the country, institutional religion such as Catholicism in contemporary China has been largely underdeveloped. Although religion has been undergoing a resurgence since the 1980s, religion in China has hardly become a unified and independent force for social transformation (Masláková and Satorová 2019). Under Xi Jinping's regime, since 2012, there has been a tightening of regulations on religious affairs, which has largely suppressed formal religious activities in China (Madsen 2020, pp. 17–33). In the meantime, the prevalence of popular and consumer culture, which partly contributes to the ongoing process of secularization, has vastly reduced the presence of religion in China's public arenas. In other words, there is little room for any religious institution to exert influence on secular Chinese public life or participate in the deliberation of public policies in China. Thus, it is unrealistic for scholars of religion in China to follow the analytical frames set by Euro-American scholars who have widely studied religion's role in the public arena, political discourse, and deliberation of public policy.

However, the invisibility of religion in secular Chinese public life does not suggest that China's religious institutions are unimportant; while they have limited influence at the societal level, they have a significant influence on ordinary religious believers and communities. We contend that CK Yang's notion of 'diffused religion' remains valid for the understanding and analysis of Chinese religion, including Chinese Catholicism (Yang 1970), which has prompted our focus on studying the lived and the socio-cultural dimensions of religion in China. Thus, using mainly ethnographic methods of data collection, and analyzing this data through the lens of lived religion, we can explore how religious individuals live according to their faith in both ordinary and extraordinary times. This can also offer us a lens through which to observe and understand the organizational structure, institutional affiliations, and individual behaviors and practices of religious belief in contemporary China.

In this article, we draw on the lived-religion approach and employ ethnographic methods to investigate religious beliefs and practices during the pandemic in a rural Chinese Catholic community (pseudonymized as Little Rome in the running text). In this paper, we reflect on the lack of attention given to religious institutions in lived-religion research, and we hope to enrich the discussion of the dynamic and diverse relations between religion and the COVID-19 pandemic.

## 3. Local Context and Methodology

The field site where this research was conducted is a rural Catholic community known as Little Rome, which is part of the Catholic Diocese of Yichang. Its political-administrative division belongs to Changyang County, Yichang Municipality, Hubei Province in Central China. According to historical records, during the 'Hundred-Year Prohibition of Christianity in China',[3] Catholics in Hubei were forced into exile in the mountainous areas of Wulin, escaping the Qing government's persecution, during which time the Little Rome community emerged.[4] With the aid of Catholic missionaries, the Catholic population

in Little Rome increased, and the community continued to develop into one of Hubei's largest Catholic communities. On 24 August 1891, the Vatican officially commissioned the Apostolic Vicariate of Southwest Hubei to the Belgian Franciscan Province of St Joseph. Under the management of Belgian priests, Little Rome gradually flourished. They built churches, opened schools, established hospitals, adopted abandoned babies, and helped refugees. These actions encouraged many non-Catholics to join the Church. According to Church statistics, 104 non-Chinese priests served in Little Rome in succession. By the time the Communists seized power in 1949, Little Rome was home to 1360 Catholics, making it the most prominent Catholic community in the Diocese of Yichang.[5]

After the founding of the People's Republic of China in 1949 and the expulsion of foreign religious ministers, the Church in Little Rome was handed over to local priests. Between the 1960s and 1970s, due to the extreme ideological antagonism, religious activities in Little Rome were forced to stop, the government confiscated the church, and the clergy were dismissed. The Church resumed operations after the 1980s, with China's implementation of the opening up and reform policies. After the long-term suppression, when religions regained their legal status in China, Little Rome's religious enthusiasm was soaring. A large number of young people chose to become priests and nuns. Since the 1980s, Little Rome has trained 38 priests and 50 nuns,[6] and it has become a well-known vocational area in China. The popularity of becoming a member of the clergy ceased around the 2000s. At this time, leaving the community to find work became a trend, and the number of people staying in Little Rome dropped drastically. Today, there are around 4000 Catholics in the Little Rome community, which is the largest Catholic gathering site in rural Hubei and a popular destination for Catholic pilgrimage in Hubei. Due to the large number of local Catholics, the diocese sent a parish priest and two nuns to serve the community.[7]

The church of Little Rome is very famous. In 1892, Hubertus Adons (Chinese name *Zanchen Huang*, 黄赞臣), a Belgian Catholic missionary, led the construction of the church and vicarage in Little Rome, which was completed in 1895. The church building remains intact today and is still in operation. The church, located at the center of the village, was constructed in the Gothic style, distinguishing itself from other local architectures as a landmark and articulating the Catholic tradition of the community.

The first author of this article entered the field site in 2015 for his doctoral research. After completing his doctoral project, he has kept close connections with the Little Rome community for follow-up studies. He continued to be updated about the Little Rome community during the pandemic via his connections and social media. After the lifting of the lockdown in Wuhan and Yichang, he visited Little Rome in October and December 2020 to conduct an ethnographic investigation into the role of the community's Catholic faith in the pandemic. The second author also participated in some aspects of the investigation and data collection.

We should note that by the time we entered Little Rome to carry out fieldwork, the pandemic had passed, and life in Little Rome had essentially returned to normal. Therefore, the vast majority of the material in this study comes from interviews. During the fieldwork, we interviewed 43 Catholics, including a priest, two nuns, and five lay co-workers in Church facilities and administrative management. In order to better understand the connections between Little Rome and the outside world during the pandemic, we also conducted telephone interviews. In the course of this study, we interviewed one priest serving abroad, three priests serving in other dioceses in Hubei Province, and six Catholics stranded in different places due to the pandemic. To ensure the accuracy of the interview data, we repeatedly sought confirmation from the interviewees via telephone and social media conversations.

This article presents the preliminary outcomes of an ethnographic study of how the Catholic faith is lived and experienced by Catholic Christians in their everyday life in the Little Rome community. It attempts to provide an in-depth account of the relationship between religion and the pandemic from the lived religion perspective. The subsequent sections illustrate the role of the Catholic faith in Little Rome and its social significance in

the time of the COVID-19 pandemic on four levels—the individual, the communal, the societal, and the trans-local.

## 4. 'We Are Children of God': Faith in Everyday Life

When asked questions regarding the practice of faith, everyday life, and their psychological state during the pandemic, members of the Little Rome community often began conversations by stating that 'we are children of God.'[8] We interpret three layers of meaning from this central motif by analyzing the interview data: (1) theologically, Catholics ought to have a personal relationship with God, which is the ultimate point of reference for their faith, construing God as a father figure in heaven; (2) communally, since all Catholics are presumed to be children of God, there is an assumed relationship of brotherhood and sisterhood between members of Catholic communities; (3) practically, as children of God, Catholics are supposed to believe the Church's doctrinal formulations, as well as complying with ecclesial disciplines and ethical teachings in their everyday life. We contend that this motif is key to our understanding of the significance of the Catholic faith for Catholic Christians whom we have studied in their everyday lives. The confessional statement 'we are children of God' lays out the theological and ideological grounding of Catholic belief and constructs the relationship between themselves and the supernatural. This construal may have provided them with immense psychological and spiritual resources with which to confront the COVID-19 crisis. Moreover, based on this interpretation, Catholics can construct a distinctive religious identity that transcends other secular identities, enabling them to go beyond social and spatial boundaries and establish connections with Catholics outside their community. Combining the rich meaning of 'we are children of God' and the situation of the Little Rome community in the context of the pandemic, we illustrate our observations from field research at four levels: the individual, the communal, the societal, and the trans-local.

### 4.1. The Individual Level: The Pandemic and the Revival of Faith

The lockdown of Wuhan City was instituted on 23 January 2020, soon followed by lockdowns of other cities, towns, and villages in Hubei Province. Most of the members of the Little Rome community who worked in Wuhan and Yichang had traveled back to the community for the Spring Festival by the time of the lockdown. For this reason, many people in Little Rome were concerned about the potential for the spread of the virus locally. The initial quarantine period was 14 days. Fear continued to grow during the quarantine.[9]

Yimeng,[10] a college student studying in Wuhan, was one of the early returnees to Little Rome. He said that he had been to many places in Wuhan with his girlfriend in early 2020, during which they may have had contact with many people. Upon returning to Little Rome, Yimeng was defined as high-risk and put under tracking and surveillance. He had to report his state of health (in particular, his body temperature) daily to local officials and medical workers. Nonetheless, Yimeng's family did not feel overly concerned about him: 'We are children of God. Everything about us comes from God. He pre-arranges everything. If he (Yimeng) was infected with this virus, it must be God's will. So, fear does not help. At that time, we prayed for him in our family every evening,' said Guilan, who is the mother of Yimeng. Apart from prayers, Yimeng's family increased their practice of other devotions, such as compline (the night prayer), devotion to the Stations of the Cross, and confession. In addition to praying at home, Guilan called the priest and offered three Masses, hoping that God would keep her family safe.

In Little Rome, the religious beliefs of individuals are closely tied to the Church. When there are significant life events, people usually go to the priest to offer Masses with particular intentions. This religiosity is manifested in Guilan's understanding of the particular role of the Church and the clergy. After Wuhan's lockdown, Father Fuxian followed the government's instructions by prohibiting local villagers' visits to the church building on all occasions. Thus, devout Catholics such as Guilan could only contact Father Fuxian to offer Mass via smartphone. 'We Catholics can only rely on God. It is not enough

to pray at home. We must go to church to offer Mass through a priest. Because the church is a holy place with the presence of the Eucharist, going to the church to offer Mass means being closer to God, and God must prefer this. We believe that the priest is very important because he knows theology and can better communicate with God. The Mass said by a priest can probably receive abundant blessing from God,' said Guilan. According to Father Fuxian, during the 76-day lockdown, 250 Masses with special intentions were held, among which only five were offered as requiem Masses for deceased relatives, and most were concerned with the pandemic. This figure was 50 times higher than that in an ordinary time. Thus, we conclude from the increase in individual devotional practices and religious rituals that there is a revival of religious enthusiasm in Little Rome due to the pandemic.

In addition to offering traditional devotional Mass services, the Church has taken advantage of internet technology to develop new ways of practicing religion, to further inspire the passion of faith in a similar way to religious organizations in other countries (Baker et al. 2020; Meza 2020). According to Nun Liu, during the pandemic, the Church continued to celebrate daily Mass in the church building without attendance. They posted photos or videos of the Mass to the Church's social media groups so that the Little Rome Catholics could all have a sense of ritual participation. At the same time, they provided some web links to encourage their members to participate in the online Mass. Nun Liu told us, 'we have moved our homilies online, which to some extent was more convenient to some people who wished to access our religious content flexibly.' The clergy intentionally shared Catholic online resources related to the pandemic, such as cases of foreign Churches organizing donations, the Pope praying for China, and Italian priests giving ventilators to the sick. They hoped to encourage the laity's faith to overcome their fear of the virus and see the power of faith in these touching events. These online activities have attracted the attention of many people, providing a rich resource for their pandemic-blighted lives and further inspiring their religious faith.

Some traditional Catholics saw the pandemic as a trial or even punishment from God to humanity, and shared many similar statements in the Church's social media groups. Later, the clergy issued stern criticism, cautioning them not to understand the pandemic in terms of eschatology-centered theological views but to see more of the power of faith and God's love for humanity. In the priest's opinion, it is important not to overplay negative emotions in such a crisis but to instead face the pandemic positively. Therefore, Father Fuxian forbade making extreme statements in social media groups.

Preceding research indicates that there is often a revival of religious faith in times of natural disasters and crises (Sibley and Bulbulia 2012; Bentzen 2019; Boguszewski et al. 2020; Gecewicz 2020). We observed the same phenomenon in Little Rome. The Catholic faith gave Little Rome spiritual support and strengthened their emotions in the fight against the virus. At the same time, the pandemic, in turn, reinforced the importance of faith and made their faith active. However, we need to emphasize that religious institutions strongly influenced the revival of faith in Little Rome. The Church provided many religious resources and developed online services to meet the needs of Little Rome's religious community as much as possible. At the same time, the clergy regulated the direction of religious revival and guided the laity to a moderate understanding of the relationship between religion and the pandemic.

During the pandemic, many people increased the frequency of their prayers. One elder villager (without recording his name, authors' note) we encountered during fieldwork said he prayed at least 100 times a day, and whenever he saw sad news, he got down on his knees and prayed. There might be exaggerations in his account; however, he is among the many villagers we met during our fieldwork who reported to have intensified their devotional practices of faith during the pandemic. After April 2020, with the nationwide containment of the pandemic, the religious fervor in Little Rome was observably lowered and returned to what has been in the ordinary time, according to the examination of our field notes and interview data. Many may pray less frequently and even become disinterested in participating in online religious activities. Father Fuxian complained:

"When you encountered a disaster, you thought of God, and when the disaster passed, you forgot about God." He admitted that this phenomenon might have been the genuine manifestation of people's religiosity, about which the clergy could do little to improve.

*4.2. The Communal Level: Identity and Communal Interaction*

Little Rome is located in the mountainous area of Wulin, a remote and economically less-well-developed region with poor infrastructure. In addition, a large number of residents in Little Rome are elderly citizens whose dwellings are geographically scattered. Therefore, the long-term lockdown has caused them enormous inconvenience. In the past, neighbors would help each other, which could solve many problems. However, in pandemic times, such actions were at significant risk. Consequently, people had to turn to the government or the Church for help.

Some members of Little Rome were migrant workers who lived in other parts of China and were unable to return home during the lockdown. Xianping, a migrant worker in Shenzhen, had to leave his mother, wife, and son behind in Little Rome. His wife and mother did not get along well with each other most of the time, so his 86-year-old mother chose to live separately in a shabby little house in the village. During the pandemic, what concerned Xianping most was his mother's well-being rather than that of his wife and son. 'They (Xianping's wife and son; authors' note) were young and well-versed with all kinds of electronic tools of communication; they could cope with life difficulties themselves. However, my mother was different. She was old, with some health conditions, and did not use a smartphone. Once in trouble, she could not notify other people for help,' Said Xianping.

In this situation, Xianping could only seek help. He first contacted the village cadres, hoping to obtain help from official channels. At that time, the number of village cadres was not sufficient, and they could only meet basic survival needs, such as helping patients travel to the hospital and helping them to buy living supplies, and there was no way to meet such a personalized request. Later, Xianping called the priest and hoped that the priest could offer some help. The priest immediately called a neighbor of Xianping's mother and asked him to take care of her and contact the Church immediately if he found any problem. In the interview, Xianping spoke about his reasons for seeking help from the Church. He knew that if he called his neighbor directly, the neighbor would most likely refuse because it was a significant risk to help someone in a crisis such as the COVID-19 pandemic. However, if the call was from the priest, it would be difficult for the neighbor to refuse out of respect for the priest. Of course, Xianping was grateful and understood this as an act in accordance with the Catholic teaching of mutual love.

Affected by the blockade policy during the pandemic, the Church could not provide direct help for some problems. At this time, the Church clergy used their particular identity to make demands of the local authority, seeking practical support for the local community. For example, there was an older woman in her 80s who lived alone in a place with no neighbors close by. Nun Liu told us that she was very concerned about the older woman and specifically called the local officials, expecting the government to offer particular care. The government immediately sent people to check the situation and solve some practical difficulties.

Apart from support among the individuals, the Church also deployed its institutional power and resources. In the pandemic, the Church in Little Rome received spiritual comfort and substantive support, such as facial masks and medical alcohol, from the government and other Catholic communities in China. The Church distributed the material goods to the local Catholic members, especially those who were commuting to work. As many people in Little Rome were laid off due to the pandemic, at the end of March, when the pandemic in Hubei was under control, the Church in Little Rome launched its reconstruction and renovation project, hiring many local unemployed workers and thus guaranteeing them basic incomes to meet their daily needs. The parish priest said the project was intended to

renovate the church building and offer opportunities to help the needy, which was in line with the Catholic moral teaching of love.

Earlier research indicates that religion facilitates communal bonding and belonging, through which members of religious communities receive comfort and support (Ellison and George 1994; Lim and Putnam 2010; Saroglou 2011). In Little Rome, the shared identity of being Catholic has strengthened the ties and connections within the local community. The testimony of the participants in our study indicates that his shared identity had a practical effect during the pandemic, generally through religious institutions. The Church used its position of authority to coordinate member contacts and solve individual problems. At the same time, it made full use of its organizational resources to provide universal help to the community whenever possible.

*4.3. The Societal Level: 'Salt and Light' in the Pandemic*

One Catholic moral teaching demands that Catholics be 'salt and light'[11] in the world, meaning that Catholics ought to participate in social affairs actively and contribute to the development of society. We observed that the Church in Little Rome has practiced what they have preached, actively cooperating with the government, employing the Church's institutional power, and constructively participating in societal efforts to contain and prevent the further spread of the pandemic.

After the pandemic outbreak, the Church was the most prominent gathering place in Little Rome, and may have constituted a conduit for spreading the virus. Therefore, the Church immediately followed the government's notice on the suspension of all activities of religious gathering at the beginning of the lockdown. According to Father Fuxian, he received phone calls and text messages from local officials of religious affairs on 24 January 2020, notifying him about the severity of the pandemic while he was celebrating a eucharistic Mass. Understanding the situation, he called off all subsequent public religious activities and asked Church members not to come to the church for 'Shanghui' (上会).[12] From 25 January 2020, the church was officially closed with a paper strip for sealing issued by the local authority. During the Spring Festival, the clergy usually visit Catholics who are sick or poor and hold religious services for Catholics living in remote areas. However, in the lockdown, the priest and nuns followed the government's instructions to stay inside the Church. In their view, they were clergy and needed to strictly adhere to the segregation policy and set a good example for Little Rome.

Although quarantined in the church, the clergy still wanted to contribute to the fight against the virus. The parish priest kept following the news on the pandemic, noting down vital statistical figures of the pandemic on a small blackboard in the church and taking pictures of the blackboard with his smartphone, sending them to the Church's social media groups to update the community. Moreover, as many older people in Little Rome were not accustomed to wearing masks and were frequently visiting the neighborhood, the priest and nuns often invoked their clerical authority to demand the elderly follow the guidelines for self-protection in the pandemic.

Moreover, the Church invoked its institutional authority and power to encourage community members to actively participate in social relief work during the pandemic. In Little Rome, many Catholics volunteered to fill in various social worker roles, such as civil vigilantes, deliverers of goods, online counselors, and communicators. The priest and nuns also employed the Church's social connections to arrange the donation of goods for medical protection and aid from other parts of China, assisting the local government in controlling the pandemic. Furthermore, the Church also donated to the government's social relief fund. The Catholic Diocese of Yichang, to which Little Rome belongs, donated 20,000 RMB during the pandemic.[13] Moreover, the Church clergy have also made several personal donations. Father Fuxian said that the local government had called him several times to express their gratitude to the Little Roman Church for its cooperation and support for their work.

During the pandemic, the Church in Little Rome took the initiative to participate in different social services and relief work, acting as a community center and hub. On the one hand, depending on the situation, the Church actively kept serving the community innovatively; on the other hand, it encouraged its members to practice religious ethics in social work, bearing the social responsibility according to the Catholic Social Teaching.[14]

*4.4. Beyond Little Rome: Faith as the Medium of Trans-Regional Connection*

Catholicism is a typical world religion in the conventional paradigm (cf. Smart 1998). Centered on the authority of the Vatican papacy, it has developed an institution of hierarchy that has spread across regional, national, ethnic, and social class boundaries all over the globe, including remote mountainous areas such as Little Rome (e.g., Madsen 1998; Lozada 2001; Harrison 2010, pp. 203–21).

In Little Rome, most Catholics are peasants who have not received much education and have remained on the margins of mainstream society. Compared with ordinary Catholics, clergy, especially priests, often interact with government officials, business people, and scholars. They have many opportunities to study, visit, and travel at home and abroad. Hence, they have high social capital and cultural capital and have many social resources. When the laity are in need, their first thought is to seek help from the clergy. In their perception, priests and nuns can and are willing to help them. An example of this perception is the case of Yadi, a Little Roman living with his pregnant wife and working for a company in Wuhan. At the beginning of the lockdown, Wuhan's social order was in chaos. Many public services, including transportation, had been suspended. When Yadi's wife was about to give birth, in much desperation and anxiety, Yadi phoned Father Fuxian for help. According to him, within less than half an hour, Father Fuxian successfully contacted a fellow Catholic in Wuhan who was willing to risk contracting the virus and agreed to drive Yadi and his wife to the hospital. 'I had no other choice but to call Father Fuxian. I knew he is well connected with many people like us in Wuhan. He is certainly our best hope.' said Yadi.

At the end of March, when the lockdown was gradually lifted, the Church in Little Rome employed its ecclesiastical networks and resources to facilitate the re-employment of its members. At that time, people from Hubei were identified as a high-risk group for infection suffering from regional discrimination. As a result, many public places were unwilling to receive people from Hubei who went back to work, which caused them a great deal of inconvenience. For example, Xiaodong was a member of the Little Rome community who worked in Tianjin, North China. He drove back to Little Rome for the Spring Festival holiday but was 'trapped' in the village due to the lockdown. At the end of March, he drove his car with a Hubei license plate back to Tianjin and could not find any hospitality services open to people of Hubei origin. In desperation, he sought help from the Church in Little Rome. The parish priest contacted the local Church in Hebei and Tianjin, which offered hospitality to Xiaodong upon his arrival.

At that time, many people in Little Rome were unemployed and had to find jobs again. Due to the economic recession, job opportunities had decreased. This, combined with discrimination against people from Hubei that arose during the pandemic, exacerbated the difficulty of finding a job. Guohua was working in a toy factory in Guangzhou. As the goods could not be exported, the boss was forced to lay off all employees, and Guohua was, unfortunately, one of them. Guohua has two children, both in junior high school, and two older people to support, and all of these expenses come from Guohua's work income. After being laid off, he could not find an ideal job, mainly because of his lower education and older age. Later, however, the priest helped him to contact a factory in Yichang run by Wenzhou Catholics, which temporarily solved his job problem.

Apart from interactions with Catholics in China, Little Rome has maintained its connections with foreign Catholic institutions. As mentioned above, there are three Chinese priests from Little Rome who are presently serving in Belgium and Australia. The Church in Little Rome has remained in contact with these outbound serving priests, with the priests

updating each other on religious and current affairs regarding the pandemic. The priests serving in Belgium contacted the Chinese Church abroad to donate some supplies to Little Rome for the fight against the pandemic. Unfortunately, this plan was not successful due to the air blockade. However, it indicates the strength of the institutional elements of the Catholic Church.

With the outbreak of the pandemic beyond China after March 2020, the Church in Little Rome was concerned about the worsening situation worldwide. They could not provide practical material help, but they constantly increased their prayers to help the Catholic Church in all parts of the world. For example, in April, when the pandemic peaked in Italy, many Catholic clergy members died, some reportedly after contracting COVID-19 during a service. So the parish priest decided to add prayers for the universal Church to their daily social media uploads. At the same time, the parish priest organized Little Roman Catholics to pray for those afflicted by the pandemic worldwide.

In general, the Little Rome community has demonstrated the positive role of faith and religion in the pandemic. On the one hand, the Catholic faith and local Church institutions strengthened community solidarity in the practice of the ethics of mutual love, care, and support among the members. On the other hand, Church institutions employed their ecclesiastical resources to provide substantive support to the community during and after the lockdown, encouraging the laities to participate in government-led efforts to contain and prevent the pandemic. They provided the laity with social resources, such as opportunities for (re)employment.

However, we must emphasize the role of Church institutions played out in the context of the social crisis brought about by the pandemic. Now, as the pandemic is under control and young people in Little Rome are going out to work, the function of Church organizations has changed again. For example, online Masses have now been canceled, and the laity once again come to church to attend Mass.

## 5. Discussion: China's Catholic Church in the Pandemic

Studies of lived religion focus excessively on individual religious beliefs and practices at the micro-level, ignoring social connections and cultural contexts at the macro-level. The case of Little Rome shows that individuals are not entirely independent but are connected to religious institutions. Moreover, in the context of a pandemic, religious institutions play a more significant role than individuals. Therefore, we strongly advocate focusing on religious institutions as part of lived-religion approaches.

In order to understand how religious institutions functioned in rural Catholic communities and influenced the everyday lives of communities during the pandemic, we must understand the social and cultural contexts in which Chinese Catholicism is embedded. Firstly, Catholicism is a highly institutionalized world religion, and religious institutions must be included in Catholic studies (Conway 2021). Little Rome is a traditional Chinese rural Catholic community that, at one point in its history, established a theocratic social order (Xiong 2018, pp. 70–73). Catholics in the community marry each other, community members and families relate to each other by bloodline and marriage, and the social structure is stable (Li 2018). The relationship between the individual and the Church can be described as "believing in belong" (cf. Day 2011), indicating the interconnection between community identity and Church membership. The Catholic hierarchy, built on the foundation of the canonical system and superimposed on the authority, order, and patriarchal morality of traditional Confucianism, gives the Church a central role in the everyday life of the community, and the clergy members enjoy a great deal of authority within the community (Madsen 1998, pp. 77–78; Harrison 2010, pp. 203–21). This particular position of the Church in Little Rome suggests that it has to assume the responsibility of supplying religious resources, empowering and enabling its social role in the pandemic.

Secondly, in the Chinese political context, religion is highly sensitive. For the unity of religions under state control, the government has implemented restrictions on religion in many respects, also granting particular favor to state-sanctioned religious institutions

based on their particular requests and demands. In exchange, religious institutions bear corresponding responsibilities and obligations, such as cooperating with the government. In the case of Little Rome, we see that the Church mediates between the state and religious laities—for example, asking the government to provide help for some Catholics who were living in difficulty. At the same time, with the Church's support, the government could better implement its policies and regulations during the pandemic, such as urging the people of Little Rome to wear facial masks.

After the outbreak of the COVID-19 pandemic, everyday life had to be centered on the containment and prevention of the virus. The Chinese government imposed strict pandemic prevention measures, and individuals had no choice but to obey the government. Social institutions, including religious institutions, were mobilized and actively participated in pandemic prevention and control efforts. During the pandemic, the influence of institutional factors on everyday life in Little Rome was important. There is no doubt that the governmental system played a dominant role. The Catholic Church can only be actively involved at the communal level, where they cooperated with the frequently renewing requirements of the government and used the Church's authority to guide Catholics to actively fight the pandemic. Therefore, to understand the religious dynamics in Little Rome during the pandemic, institutional factors such as the role of the Church need to be considered.

Lastly, we are aware of and need to address the issue of ostensible clericalism in our presentation of the institutions of the Church in Little Rome, as Father Fuxian was often represented as the voice of the Church. In our fieldwork, we observed that the Catholics in Little Rome viewed the priest as 'the representative of the Church, just as Jesus is the representative of God on earth.' In other words, the parish priest in Little Rome is regarded as both a theological and managerial authority by the laity. The emphasis on the priestly authority is also embedded in the patriarchal social order in traditional Chinese culture (Madsen 1998, p. 136). In recent years, by order of the government and the diocese, two official religious institutions, the Little Rome Catholic Patriotic Association and the Little Rome Catholic Management Committee were established and headed by Father Fuxian. To better operate the two institutions, the nuns and the laity were involved in the management team. The management of the two institutions submitted to the authority of Father Fuxian in terms of theological authority; however, in practice, the nuns and the laity also had the power in decision-making. In addition, apart from being responsible for the Church management, Father Fuxian has been the person held accountable to the government on public religious affairs in Little Rome (see in the above example where he was informed by the local official regarding issues regarding the pandemic). Thus, instead of identifying the role of Father Fuxian, the parish priest in Little Rome, with clericalism, we see him as the legitimate institutional representative of the Church in Little Rome.

In addition, we would like to briefly articulate the contribution of this study. Existing studies of lived religion have focused on religion in European and American societies; researchers promote personal religious experience and practice, focusing on resistance to institutional religion. When we extend living religion to other cultures, countries, and social settings, such as rural Chinese Catholicism, we see a different presentation and structure of lived religion, and this will expand the scope and depth of the study of lived religion (Ammerman 2016, 2020). This can also help us to capture the process of religion and the pandemic in different places and explore the diversity of the relationship between religion and the pandemic.

## 6. Conclusions

This article presents a local case study of a rural Catholic community in China during the COVID-19 pandemic. It provides an ethnographic account of how the religious institution and individual members practice the Catholic faith at the individual, communal, societal, and trans-local levels. Confronting the pandemic, the Chinese government forcefully took a top-down approach, instituting measures and policies to implement an

unprecedented nationwide lockdown, to which Little Rome was no exception. Nevertheless, the case of Little Rome deserves to be given further attention due to its predominant Catholic culture and its proximity to Wuhan, the initial epicenter of the pandemic. Moreover, we believe that the case of Little Rome provides an ideal opportunity for us to observe the role of religion, Catholicism in particular, in China's confrontation with the pandemic.

Since most people in Little Rome are traditional Catholics, the belief that "we are God's children" is an essential spiritual and psychological resource and affects their faith practices and social actions. Through the lens of lived religion, the religious motif of 'we are children of God' can reveal to us the multi-faceted relationship between belief, emotion, practice, interactions between individuals and institutions, and the blurred boundaries between individuals, families, communities, regions, and nations. In this process, the Church institutions in Little Rome played an important role as a bridge between individuals and society, families and communities, and the local and global.

In the case of Little Rome, we can see the importance of religious institutions in the everyday life of rural Chinese Catholics. Therefore, we argue that religious institutions should be included in lived-religion studies and that attention should be paid to lived religion in different places, countries, and cultures, which could complement existing studies on lived religion and expand the in-depth development of the lived-religion approach.

We argue that through in-depth ethnographic inquiry, the people whose lives are centered on the particular locality and culture of Little Rome have provided us with rich data with which to understand the dynamic and diverse relationships between religion and the pandemic. As vaccination programs have been introduced in many countries, the world is beginning to recover from the shadow of the pandemic. As scholars of religion, we contend that more in-depth investigations of this sort will be possible as lockdown policies are lifted, and such studies are helpful in the effort to document and understand the dynamics of religion in the pandemic.

**Author Contributions:** Conceptualization, W.X.; data curation, W.X. and X.L.; writing—original draft preparation, W.X.; writing—review and editing, X.L.; project administration, W.X.; funding acquisition, W.X. All authors have read and agreed to the published version of the manuscript.

**Funding:** This research received no external funding.

**Data Availability Statement:** All data in this study were obtained from our fieldwork.

**Conflicts of Interest:** The author declares no conflict of interest.

## Notes

1. This information comes from WHO. Available: https://www.who.int/director-general/speeches/detail/who-director-general-s-opening-remarks-at-the-media-briefing-on-covid-19---11-march-2020, accessed on 1 March 2021.
2. Pew Research Center. 2020. Most Americans say coronavirus outbreak has impacted their lives. March 30. Available: https://www.pewresearch.org/social-trends/2020/03/30/most-americans-say-coronavirus-outbreak-has-impacted-their-lives/, accessed on 15 June 2021.
3. The 'Hundred-Year Prohibition of Christianity in China' was related to the famous 'Rite Controversy' during the mid-Qing Dynasty. For details, see (Bays 2011, pp. 66–91).
4. The United Front Work Department of Changyang County, History of Catholic Mission in Shiguntang (manuscript), Changyang County Archives, 1964: 3. [中共长阳县委统战部, 天主教在石滚淌的传教史 (手稿本), 长阳县档案馆藏, 1964: 3.] The United Front Work Department of Changyang County, the History of Changyang Catholic Church of the Catholic Diocese of Yichang (manuscript), Changyang County Archives, 1965: 5–6. [中共长阳县委统战部, 天主教宜昌教区长阳天主堂史志 (手稿本), 长阳县档案馆藏, 1965: 5–6.]
5. The United Front Work Department of Changyang County, the History of Changyang Catholic Church of the Catholic Diocese of Yichang (manuscript), Changyang County Archives, 1965: 83. [中共长阳县委统战部, 天主教宜昌教区长阳天主堂史志 (手稿本), 长阳县档案馆藏, 1965: 83.]
6. These are historical data, including priests and nuns who have passed away or left the church clergy. Currently, there are 30 priests and four nuns. At present, they are serving in Hubei, Beijing, Henan, and Shanxi; two are serving in Australia; and one is serving in Belgium.
7. For a detailed background of the Little Rome community, see (Xiong 2018, pp. 53–79).

8    This motif is referred to the Christian Bible in 1-John 3:1 (RSV), which is a typical Christian parlance in the everyday life of little Romans.

9    Fortunately, the county where Little Rome was located remained safe, with zero confirmed cases during the pandemic.

10   All names in the article are pseudonyms for the sake of confidentiality.

11   This motif is referring to Matthew 5:13–16 in the Christian Bible (RSV).

12   'Shanghui', meaningfully rendered as 'General Assembly', is a common practice in Chinese rural Catholicism. Due to the shortage of Catholic clerics in rural China, few rural Chinese Catholic churches have long-term parish priests. In important Chinese festive seasons such as the Spring Festival, priests can travel to rural and mountainous areas where Catholic communities are located to celebrate mass and hold catechetical sessions and other religious ceremonies. This is a tradition established by Western Catholic missionaries and kept alive up to the present. As explained in the main text, people are scattered in Little Rome due to its complicated mountainous geography. The tradition of 'Shanghui' is therefore of great significance for Catholics in rural Little Rome, as it meets the religious needs of most local Catholics. 'Shanghui' usually lasts seven days and is the largest collective event during the Spring Festival in Little Rome.

13   This information was provided by Father Fuxian.

14   For more information about the Catholic Social Teaching, see: http://www.vatican.va/roman_curia/pontifical_councils/justpeace/documents/rc_pc_justpeace_doc_20060526_compendio-dott-soc_en.html, accessed on 1 March 2021.

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
