# Peer review of "‘We Are Children of God’: An Ethnography of a Catholic Community in Rural China in the COVID-19 Pandemic"

_religions, doi:10.3390/rel12060448_

Round 1
Reviewer 1 Report
The paper is interesting and covers a new and unexplored topic. The point of view of the Catholics of a Chinese village in facing the difficulties of the pandemic is explored, setting it in the wider context of the role of religions in times of crisis. The four levels explored are finally well presented and developed. The language is clear and delivers the message fluently.
The issue is of certain interest at the moment and is hoped to be further explored in the future.
It is suggested to add the Bible or Gospel reference to the quotations "We are children of God" and "salt and light".
Author Response
Thank you for your comments, we have revised the manuscript according to your suggestions. Please see the attachment.

Reviewer 2 Report
This paper presents how Catholics from a rural village in China have responded to the COVID-19 crisis. Elaborating on 43 individual interviews, and engaging with a set of theories known as lived religion, the authors argue that “the Church” gained renewed importance during the pandemic – and that research on lived religion should pay more attention to religious institutions.
This research is extremely interesting and timely. With COVID-19, doing fieldwork in mainland China has become increasingly difficult, and little is known about the ways Chinese Catholics responded to the pandemic. Furthermore, this paper engaged with current theoretical debates in religious studies and seeks to apply notions surrounding ‘lived religion’ in its investigation.
In this regard, the first paradox of the paper is about its methodology. While the authors apply ‘lived religion’ as a theoretical framework, they seem to rely mostly on interviews without given us a sense of the daily religious life of those Catholics. The authors went to the field site, but nothing is said about what they could have observed and witnessed there. Thus, the authors seem to limit themselves to what people say. Consequently, the paper gives the impression that ‘lived religion’ becomes a ‘discursive/narrated religion’. It might be interesting to describe – and discuss – the lived religion that Chinese Catholics from this village have performed during the pandemic. Which kind of prayers did they say? Which kind of daily/weekly rituals did they perform? How has that evolved over time? In this regard, I would recommend the excellent series of articles published by the Asia Research Center on Religion and Covid-19 in Asia (CoronAsur).
A second paradox revolves around the notion of “the church as an institution”. While the authors emphasize the importance of this actor in the lived reality of rural Catholicism during COVID-19, it seems that “the Church” is indeed Father Fuxian, the parish priest of this local community, and his extended network. So, my question is: what is the Church – what is an institution? Thus, I would recommend adding an analytical discussion of the notion of the Church seems it seems to be the main finding of this research. What is the Church that rural Catholics are referring to? If the church is a network of relationships, how is that echoing what Mayfair Yang and Yan Yunxiang have debated about guanxi in Chinese society? If the Church is the clergy, which kind of clergy are we talking about? Is it the priest only – which seems a very 19th century Western Catholic model and therefore, something quite new in China— or does it also include the “nuns” as mentioned in the article? Also, how is this intricate relation between the clergy and the Church responding to theories on Chinese Christianity as we can read in “Making Christ Present in China” (Palgrave 2020)?
Finally, the paper still has a series of grammatical and typo problems (L.246,250,261, etc.). Thus, additional editing would be best.
Author Response

(The authors gave the same response as above.)

Round 2
Reviewer 2 Report
Most of the comments and suggestions have been addressed. The authors have responded to our concerns on distinguishing clergy and the Church while clarifying a few additional points. To nuanced L. 59, I would still recommend "The Pandemic: Perspectives on Asia” edited by Vinayak Chaturvedi (as well as Coronasur). Overall, the contribution of the article is to highlight the importance of clergy members and institutions in lived religion - an aspect that this theoretical and methodological approach tends to underestimate. Yet, the article remains week in several aspects. On the theoretical aspect, little is said about what 'institution' really means - and how clergy and institution intersect and differentiate from each other. On the methodological aspect, and due to the circumstances of the investigation (the pandemic), the research has been mostly limited to interviews (which is a contradiction with the theoretical engagement of this paper with lived religion). Nevertheless, the strength of this paper is certainly to provide fresh and nuanced data on the life of Chinese Catholics during COVID-19. It is a very timely contribution that needs to be published as soon as possible.
This manuscript is a resubmission of an earlier submission. The following is a list of the peer review reports and author responses from that submission.
Round 1
Reviewer 1 Report
This is an interesting, well-written and worthwhile article. The presented ethnographic research brings important data about the Catholic Church in China during the pandemic.
I would only recommend the authors to add a chapter on methodology - how many participants were interviewed, when, what is their status within the church?
Author Response
Thanks for your review and the comments are very helpful, it played a key role in the revision of our manuscript.Please see the attachment.

Reviewer 2 Report
The subject explored in the article under review is timely and fits well with the focus of the special issue. The relationship between religion and the Covid-19 pandemic is explored with reference to a particular local context and it is studied at four levels: individual, communal, societal, and trans-local.
The manuscript has not been sufficiently “blinded” for the purpose of the review, with these statements appearing in the “Local Context and Methodology” section: “Wei Xiong, the leading researcher of this article…”; “Xinan Li, another author of this paper, also participated in part of the investigation and data collection…”
The subheadings do not correspond with the content very well. For example, in the section with the heading “The Individual Level: Natural Disaster and the Revival of Faith”, there is no discussion on natural disasters, the term is used once in reference to other research.
The authors provide no details regarding the recruitment of participants, the sample and the methods of data analysis. These should be added to the manuscript. How many people did you spoke to? How did you chose your respondents? How did you analyse the data/ chose the quotes presented in the paper?
There are major problems with English. The paper requires extensive revisions in this respect. In the current version of the manuscript many sentences as difficult to understand (e.g. “This concern has led to some the reluctance to comply with, even 26 negative reactions to the state policies under the pandemic, which increases the risk of the 27 wide spread of virus (DeFranza 2020).”), grammatically incorrect (e.g. “In the West, there sees a resurgence of religion in the public arena (Reynolds 2015)”; “People’s emotion of fear reaches to the peak during the time of quarantine. We observe that with time goes by, the emotion of fear and panic in Little Rome gradually eased.”), or confusing (e.g. “Soon after Wuhan’s lockdown, people in Little Rome instantly became nervous…” – was it “soon after”, or “instantly”?), many colocations used in the text do not appear in English. The forms of verbs do not match plural/singular subject (e.g. “In Netherland, researchers finds 30 that religion is in part responsible for…”). In some places wrong words are used (e.g. in the sentence “Guihua, one of our interviewees, expressed her emotion of anxiety when 151 memorising times early in the pandemic” memorising is not a correct word). The revision of language should be thorough and applied to the whole text, not just the examples given above (this includes also the translation of respondents’ narratives).
The authors declare focus on the lived religion, but in their discussion give lots of attention to the local Church, its position and role in securing supplies during the pandemic, institutional interpretations of the pandemic, institutional religious practices and teachings. While any discussion of lived religion should reference the institutional religion as well, it would be good, if the relationship between the two was discussed and explored in more depth in the particular context presented in the paper.
The presentation of data, while very interesting for the reader unfamiliar with the details of the pandemic outbreak and management in the region and with the religious community of Little Rome, is purely descriptive. Given the ethnographic approach this is justified, however I think the manuscript would benefit greatly from stronger linking the discussion to some theoretical concepts. These would make it easier to compare the case presented in the paper with other cases and draw broader conclusions about the relationship between religion and pandemic. Even the framework of lived religion, explicitly identified by the authors as their approach to studying the religion in the context of the pandemic, seems to disappear in the presentation of the empirical material. The references to other studies are also very sparse in the discussion of the empirical material.
In the concluding section of the paragraph on the role of the Church in providing practical help to residents of Little Rome during pandemic, the authors state: “There has been some literature pointing out that different from Protestantism which emphasis individual responsibility, Catholicism has always encouraged the notion of social responsibility (Colzato et al 2010).” I would argue that this is normative statement which introduces unnecessary bias. The community orientated work by Protestant Churches, including many charity initiatives is well documented in the literature.
Author Response

(The authors gave the same response as above.)

Reviewer 3 Report
The paper is interesting and covers a new and unexplored topic. The point of view of the Catholics of a Chinese village in facing the difficulties of the pandemic is explored, setting it in the wider context of the role of religions in times of crisis. The four levels explored are well presented, despite not so thoroughly developed. After a brief description of the village examined, the author presents the reaction to the spread of pandemic and to lockdown examining the life of Chinese Catholics firstly on the individual level, secondly, on the communal level, thirdly on the societal level, and fourthly on the trans-local level. This structure is interesting and can be used as a model for further analysis of the interaction between the pandemic and religions in other settings.
The topic proves to be interesting and seems to require further analysis, in particular, some suggestions are given below to improve the paper:
- further explore the four levels of analysis giving also a general view before presenting the individual cases
- present the situation and features of the village in the years 1891-1949 and after the 1980s up to now (lines 82-84 and after 89)
- make direct reference to the fourth level mentioned in lines 115-116, the trans-local, in paragraph 3.4. The reference to other Catholic Chinese communities, both in China or abroad, would better support the analysis.
In general, the fluency of the English language is not perceived. In many occurrences verbs and prepositions are used in the wrong way and this prevents an enjoyable reading (i.e. lines 9 “this” instead of “these”; 39 “turn” instead of “turned”; 78 “was” instead of “were”; 127 “are” instead of “is”; 454 “ethnic” instead of “ethic”). A copyediting is suggested.
I suppose that one of the authors of the 2020 paper in line 525, Mariapaola Barbato, has Mariapaola as name and Barbato as surname, so the reference in line 44 should be “Thomas and Barbato”.
Author Response

(The authors gave the same response as above.)

Round 2
Reviewer 2 Report
Thank you for the work you put into revising the manuscript. This is much imporved, the argumentation is clearer and more consistent. The language is much better, although, there are still occasional probelms with English (e.g. . In addition, the church clergies made several donations by personal).
I congratulate you on your timely and original work.